# Noise-Induced Hearing Loss

**DOI:** 10.3390/jcm12062347

**Published:** 2023-03-17

**Authors:** Nirvikalpa Natarajan, Shelley Batts, Konstantina M. Stankovic

**Affiliations:** 1Department of Otolaryngology-Head and Neck Surgery, Stanford University School of Medicine, Palo Alto, CA 94304, USA; nn9@stanford.edu (N.N.); sbatts@stanford.edu (S.B.); 2Department of Neurosurgery, Stanford University School of Medicine, Palo Alto, CA 94304, USA; 3Wu Tsai Neuroscience Institute, Stanford University, Stanford, CA 94305, USA

**Keywords:** noise-induced hearing loss, sensorineural hearing loss, cochlear hair cell, diagnosis, prevention, screening, review

## Abstract

Noise-induced hearing loss (NIHL) is the second most common cause of sensorineural hearing loss, after age-related hearing loss, and affects approximately 5% of the world’s population. NIHL is associated with substantial physical, mental, social, and economic impacts at the patient and societal levels. Stress and social isolation in patients’ workplace and personal lives contribute to quality-of-life decrements which may often go undetected. The pathophysiology of NIHL is multifactorial and complex, encompassing genetic and environmental factors with substantial occupational contributions. The diagnosis and screening of NIHL are conducted by reviewing a patient’s history of noise exposure, audiograms, speech-in-noise test results, and measurements of distortion product otoacoustic emissions and auditory brainstem response. Essential aspects of decreasing the burden of NIHL are prevention and early detection, such as implementation of educational and screening programs in routine primary care and specialty clinics. Additionally, current research on the pharmacological treatment of NIHL includes anti-inflammatory, antioxidant, anti-excitatory, and anti-apoptotic agents. Although there have been substantial advances in understanding the pathophysiology of NIHL, there remain low levels of evidence for effective pharmacotherapeutic interventions. Future directions should include personalized prevention and targeted treatment strategies based on a holistic view of an individual’s occupation, genetics, and pathology.

## 1. Introduction

Noise-induced hearing loss (NIHL) is a consequence of multifactorial damage to auditory structures following exposure to occupational, environmental, or recreational sources of loud sound. Noise has been recognized as a factor contributing to hearing loss long before rigorous data collection, sophisticated analyses, and careful experimental design became the norm. Although earplugs were patented in 1864, hearing protection devices are mentioned in ancient Greek mythology [1]. NIHL was formally acknowledged as a medical condition in the United States (US) during the Industrial Revolution, first named ‘boilermaker’s disease’ as a reference to the hearing loss suffered by workers building engines that powered transportation and production [2]. Historical data on US women who worked in the factories during World War I and II reveal devastating health effects, including hearing loss, although disorders caused by exposure to chemicals received more attention than those attributable to noise [3]. Noted physician and Nobel Prize winner Robert Koch predicted in 1910 that “one day man will have to fight noise as fiercely as cholera and pest” [4]. Despite this prediction and the long-standing knowledge of the adverse effects of noise on hearing and extensive research in the modern era, hearing loss continues to rank among the most common work-related illnesses both in the US and the world [5].

NIHL may be unilateral (affecting one ear) or bilateral (affecting both ears), and the hearing deficits may be transient or permanent [6]. The duration and severity of NIHL depends on the extent and location of cellular damage, which correlates with intensity and duration of the sound stimulus. Because the mammalian auditory sensory epithelium—the organ of Corti—does not spontaneously regenerate when sensory cells are lost, noise-induced hair cell or neural degeneration can result in permanent hearing loss particularly in the setting of repeated exposure [6,7]. Furthermore, NIHL is frequently irreversible and can have a profoundly negative impact on an individual’s quality of life and on the economy and society at large. However, NIHL is largely a preventable condition when appropriate precautions, such as the use of hearing protection, can be taken. Therefore, implementing measures to detect and attenuate causative factors, raising awareness of the condition and implementing protective strategies, and developing therapies that protect against or mitigate damage from noise exposure can aid in the prevention of this common condition. In this review, we describe the epidemiology, anatomy, pathophysiology, diagnosis, and prevention strategies of NIHL and protective pharmacological agents which have demonstrated some efficacy in humans, and we close with an outlook on emerging therapies.

## 2. Epidemiology

The most recent Global Burden of Disease report (2019) estimated that 1.57 billion people, or 20.3% of the world population, are affected by any kind of hearing loss, with 62% over the age of 50 years [8]. As NIHL is the second most common cause of hearing loss after presbycusis (age-related hearing loss) [9,10], it imposes an enormous burden on individuals and health systems. Globally, NIHL is estimated to affect approximately 5% of the population and is generally more common among adult men [11,12]. However, this may be an underestimate as the prevalence of NIHL varies widely across populations and age groups. For example, greater exposure to occupational and urban noise in developing nations increases the risk of NIHL, and limited access to healthcare and screening tests may leave much of the burden undetected [13,14,15]. Further, developing nations may lack governmental guidance or legislation to limit noise exposure or lack public education measures to encourage the use of hearing protection. This can be observed in data on the prevalence of occupational NIHL across nations within the same geographic region. Approximately 16% of disabling hearing loss in adults globally is attributed to occupational noise, ranging from 7 to 21% across various geographic regions [16]. Both the highest and lowest rates are in the West Pacific region (as defined by the World Health Organization (WHO)): rates are lowest among developed nations including Australia, Japan, New Zealand, and Singapore, and the highest in the region include Cambodia, Laos, Malaysia, Philippines, and Vietnam [16].

Due to lack of public records and funding for research on NIHL, the true burden of NIHL in the developing world is unknown, although several studies in India and Africa have highlighted it [17,18]. For example, a survey of the effects of urban noise on traffic police in Hyderabad, India, reported that 76% had NIHL, and high rates of NIHL were observed in workers in textile, mining, and heavy engineering industries [17]. Similarly, a cross-sectional study reported that the prevalence of NIHL among Tanzanian iron/steel workers was significantly higher than among schoolteachers (48% vs. 31%) but was high in both groups [19].

NIHL is also common among developed nations. In the US, interviews and hearing tests conducted by the Centers for Disease Control and Prevention (CDC) National Health and Nutrition Examination Survey (NHANES) led to estimates that at least 6% and up to 24% of adults have hearing loss in one or both ears due to noise exposure, with higher rates among males [20,21]. An estimated 17% of US youths (aged 12–19 years) have hearing tests indicative of NIHL, with higher rates among females, primarily attributable to unsafe recreational noise levels [22]. In the European Union, noise is a main cause of disabling hearing loss, affecting over 34.4 million people in 2019 and contributing to over 185 billion euros in annual costs related to reduced quality of life and lost productivity [23]. NIHL is also reported to be more prevalent in eastern and central Europe compared to western Europe [24]. In the United Kingdom (UK), approximately 180,000 people aged 35–64 years were estimated to have severe hearing difficulties attributable to noise at work in 2002 [25]. NIHL and occupational deafness have been declining in the UK but are still estimated to impact over 11,000 workers annually [26].

## 3. Pathophysiology

### 3.1. Harmful Noise Levels

Sound frequency is measured in Hertz (Hz), and intensity (loudness) is measured by sound pressure level on a logarithmic decibel (dB) scale, which ranges from safe to unsafe exposure levels (Figure 1). Normal human sound discrimination typically begins at 0 dB within the frequency ranges of 20 Hz to 20,000 kHz [27]. In comparison, a normal conversation is approximately 60 dB, traffic is 80 dB, very loud music at a rock concert or nightclub is 120 dB, and a jet engine is 140 dB. Broad categories of noise exposure include continuous noise, which is sustained over time, and impulsive noise, which occurs rapidly (i.e., a gunshot or explosion) and is generally at a higher sound pressure level [28]. A sound pressure level above 110 dB is considered discomfort threshold, and above 130 dB is the pain threshold [3]. Sustained noise above 70 dB can result in cumulative hearing loss while noise above 120 dB can cause immediate hearing loss [29].

### 3.2. Auditory Structures and Functions Impacted by Noise

#### 3.2.1. Sound Transmission from the Outer to the Middle and Inner Ear

The damage associated with NIHL begins when harmful sound is channeled to auditory structures from the outer ear via the auditory canal (Figure 2a). The unique shape of the auditory canal, where it is closed at one end by the tympanic membrane and open at the lateral end, serves as a quarter-wave resonator tube funneling sound to the middle ear [32]. Sound waves vibrate the tympanic membrane which transmits information about the frequency and amplitude of sound through the air-filled middle ear cavity via the tiny ossicles (malleus, incus, and finally to the stapes) [33]. Amplification of sound pressure is a key feature of the transmission of sound from the middle to inner ear, allowing the stapes push against the higher resistance fluid in the inner ear behind the oval window [34,35,36]. As a result of the larger surface area of the tympanic membrane compared to the cochlea’s oval window (a ratio of 20:1), the sound pressure ultimately received by the oval window is approximately 20 times greater than the original sound stimulus [37]. Resonance in the ear canal produces amplification of acoustic frequencies whose wavelengths are approximately four times the length of the canal, which, in humans, results in enhancement of frequencies around 4 kHz [38].

The mammalian cochlea is a spiral-shaped structure divided into three fluid-filled chambers—the scala tympani, the scala vestibuli, and the scala media (i.e., the cochlear duct) (Figure 2b). In humans, the cochlea has a conical central canal (modiolus) around which the cochlear duct makes two- and three-quarter turns. The footplate of the stapes rests on the cochlea’s oval window and moves in a piston-like manner to transmit sound vibrations to the cochlear duct commensurate with the loudness and intensity of the sound stimulus. Accordingly, vibrations from displacement of the oval window spiral up the cochlea to cause physical movement of fluids and of the flexible basilar membrane. The basilar membrane does not vibrate as a whole; rather, specific areas vibrate in response to different sound frequencies. This is achieved by variations in stiffness and width along its length: it is thin and stiff at the base and broader and more flexible toward the apex. Therefore, lower frequencies vibrate the basilar membrane closer to the apex of the cochlea while higher frequencies produce vibrations closer to the base, near the oval window. This arrangement is known as a tonotopic organization [39,40]. Thus, our ability to differentiate sounds of varying loudness and pitch depends on the ability of the cochlea to respond appropriately to transmitted sound amplitudes and frequencies.

#### 3.2.2. Auditory Transduction in the Organ of Corti

The cochlea’s scala media contains the organ of Corti, the cellular apparatus for transduction of sound vibration into neural signals that can be interpreted by the brain. The organ of Corti is a mosaic of sensory hair cells and non-sensory supporting cells which, in mammals, does not have the capacity to replace lost cells once damaged (Figure 2c). There are two types of sensory hair cells: a single row of inner hair cells (IHC; ~3500 in humans) and three to five rows of outer hair cells (OHC; ~12,000) [41]. Both types of hair cells have 3–4 rows of actin-rich “hairs” called stereocilia, which increase in height along the basilar membrane from base to apex. The roles of the supporting cells are to generally provide structural, metabolic, and immune support to the organ of Corti and are essential for hair cell survival. These include pillar cells, separating IHC and OHC to form the tunnel of Corti, as well as Deiters’, Hensen’s, and Claudius cells [42].

While the scala vestibuli communicates with the scala tympani at the apex of the cochlea (helicotrema), the scala media is separated from both chambers by Reissner’s membrane and the basilar membrane, respectively. This separation is crucial for the maintenance of the unique ionic composition of the extracellular fluids in these chambers, generating the endocochlear potential which is essential for auditory transduction. Specifically, the scala vestibuli and scala tympani contain perilymph which is K+-poor and Na+-rich. Conversely, the scala media is filled with K+-rich and Na+-poor endolymph which is secreted by a specialized epithelium called the stria vascularis. The tips of the hair cells’ stereocilia are embedded in the flexible tectorial membrane and move in response to sound vibrations. The stereocilia bundles are also connected to each other (within the same hair cell) via extracellular tip links formed in part by cadherin 23 (CDH23) and protocadherin 15 (PCDH15) proteins [43]. These links allow the stereocilia to move together and cause non-selective mechanically gated ion channels to open, conducting inward currents of cations (K^+^ and Ca^2+^) from endolymph [44].

While the endocochlear potential of endolymph in the scala media is +80 mV compared to perilymph [45], hair cells have a potential of −55 mV to up to −150 to mV [46]. Thus, the influx of cations depolarizes the hair cell, resulting in different effects in OHCs and IHCs. OHCs physically elongate and contract, activating the “cochlear amplifier” to narrow the area of excitation and give the traveling sound waves more sensitivity [47,48]. This aids in frequency discrimination and in the detection of quiet sounds. Meanwhile, IHCs release the neurotransmitter glutamate to synaptic boutons of type I (myelinated) spiral ganglion neurites (SGNs) which form the auditory nerve. Glutamate activates the AMPA glutamate receptors in the afferent nerve fibers which in turn transmit the neural signal to the cochlear nucleus and higher auditory processing regions. A single IHC synapses with approximately 10 boutons from 10 different spiral ganglion cells, but the number varies tonotopically (can be up to 20 boutons in the most sensitive frequency regions). IHCs have synaptic ribbons which sustain high rates of glutamate release and maintain synchronous auditory signaling [49,50,51,52]. The temporal change in neuronal firing is rapid in the ear, to the effect of 2 to 5 kHz, and distinctions between these various frequencies facilitates sound discrimination [51,53].

### 3.3. Mechanisms of Damage in NIHL

The mechanisms by which loud noise induces hearing loss include mechanical damage of cochlear structures, reduction in blood flow, sterile inflammation, and oxidative stress and excitotoxicity due to overstimulation of hair cells and nerves (Figure 3). The loss of hair cells via apoptosis is ultimately the most severe injury and contributes to permanent hearing loss [54,55]. Noise at specific frequencies can cause discrete areas of hair cell damage which manifest as frequency-specific hearing deficits [56,57].

The variables impacting the severity of damage include factors attributable to the sound stimulus (i.e., intensity, spectral energy of noise exposure, and duration) and the physical, mechanical, and chemical characteristics of the outer, middle, and inner ear. The damage caused by noise can cause temporary or permanent damage and associated hearing loss. Prolonged duration of exposure to hazardous noise levels or a one-time exposure to high-intensity sound levels can both cause permanent threshold shifts (PTS) [6,38,58]. PTS occurs when the standard threshold is stabilized at an elevated level due to the destruction of the cochlear hair cells, mechanosensory hair bundles, or nerve fibers [34,59]. Hearing loss persisting at 14 days after noise exposure, with the upper recovery limit being 30 days, is indicative of PTS [58]. Transient attenuation of hearing with recovery within 24–48 h is called a temporary threshold shift (TTS) and results from more moderate noise damage [60]. TTS and PTS show distinct histopathological patterns, as described below.

#### 3.3.1. Mechanical Damage

Because the hair cells sit atop the basilar membrane and because their stereocilia are embedded in the tectorial membrane, they are subject to mechanical shearing forces in response to sound vibrations. This continuous mechanical stress causes damage over time and is a feature of typical age-related hearing loss. However, in the context of intense or persistent noise, the shearing forces can cause stereocilia core breakage, destruction of tip links, and ultimately premature hair cell death [61]. At low frequencies of 2 kHz and below, the middle ear muscle reflexes may contract on exposure to noise and provide a degree of protection against mechanical shearing forces [62]. Non-linearities in these frequencies, auditory responses, and the mechanical response differences in the basilar membrane explain the finding that low-frequency hearing thresholds are less vulnerable to the loss of apical OHCs [63,64].

Loss of stereocilia tip links due to excessive force, uncoupling the mechanoelectrical transduction (MET) channel, precludes mechanotransduction by the hair cell [65]. However, the tip links have some capacity to repair via replacement of PCDH15 and CDH23 proteins. In mammals, tip links broken in vitro have been observed to repair within 24 h, followed by restoration of mechanotransduction, although the MET current remains impaired for 36 h [66,67]. This phenomenon is suggested to underlie TTS after noise exposure, although PTS may occur if the tip links are damaged too extensively to repair. Furthermore, noise may damage the F-actin core of the stereocilia itself. Mechanical overstimulation can decrease the stiffness of the core, leaving stereocilia with a “floppy” appearance resulting from actin depolymerization, loss of actin cross-links, or fusion with other stereocilia [68]. As the stereocilia have some capacity to repair actin, this can result in TTS or PTS depending on the extent of the damage to the stereocilia core [61,68].

Supporting cells may also be physically damaged by noise. Pillar cell damage has been observed both after high-level impulse (160 dB) and continuous noise (100–120 dB SPL) [69,70]. Dieters’ and Hensen’s cells also have a protective effect on OHCs, and acoustic trauma may displace them toward the center of the cochlear turn, resulting in loss of hearing sensitivity [71]. Buckling of the supporting cells, particularly the pillar cells, has been shown to result in an uncoupling of the OHC stereocilia from the tectorial membrane in a chinchilla model of NIHL [72]. This decreases the hair cell stimulation and results in a TTS. Moderate noise exposure can also result in microchemical changes, decreasing pillar rigidity [72].

#### 3.3.2. Oxidative Stress and Reduced Blood Flow

Apart from their function in the Krebs cycle, the mitochondria play an essential role in apoptosis and oxidative stress. Following infection, trauma, and in this context, noise-induced trauma, cellular damage or degradation of cell connections may lead to the activation of apoptotic pathways by hair cells [73]. Reactive oxygen species (ROS, e.g., superoxide, hydroxyl ion, hydrogen peroxide) may acquire an unstable number of electrons which also triggers pro-apoptotic pathways [73,74]. On receiving apoptotic signals, the balance between pro-apoptotic proteins, such as BAX or BAD, and anti-apoptotic proteins, such as those in the BCL family, is tilted [75]. Outer mitochondrial membrane permeability increases, leading to the transport of cytochrome c from the inner mitochondrial membrane to the cytoplasm [76]. Caspase enzymes in the cytoplasm are activated, resulting in cell degradation [76]. Yamane et al. demonstrated the stagnation of blood flow and the disturbance of strial circulation following acoustic trauma [77] and the appearance of free radicals such as the superoxide ion in the luminal surface of the stria vascularis marginal cells [78]. Stress and ischemia activate MAP3 kinases which in turn activate the JNK pathway, and phosphorylated JNK enters the nucleus and activates pro-apoptotic pathways such as c-Jun and Fos [79]. The downstream products lead to the expression of transcription factors and cellular effectors of apoptosis [76].

Peroxynitrite (ONOO–), a highly reactive nitrogen species (RNS) derived from NO and superoxide, is one of the most harmful free radicals to hair cells [80]. Noise-induced generation of RNS and ROS continues as an active biochemical process, not just in the immediate aftermath of exposure, but progresses over more than ten days post-exposure and can cause subsequent damage to cochlear structures [81]. Antioxidant enzymes such as superoxide dismutase play an important role in countering free radicals by catalyzing the dismutation of superoxide to hydrogen peroxide [82]. Other enzymes such as glutathione, glutathione peroxide, and catalase then balance with *Sod1* to provide cellular defense, and the loss of *Sod1* in knockout mice increased the susceptibility to noise-induced PTS [83].

The stria vascularis forms the lateral wall of the cochlear duct and comprises three cell layers: marginal, intermediate, and basal. It is responsible for maintaining the endocochlear potential and has a rich blood supply [84,85]. Hazardous noise exposure can cause damage, particularly to the intermediate cells, leading to temporary or permanent changes in the endocochlear potential and impaired hair cell mechanotransduction [86,87]. Furthermore, disruption of blood flow may induce cell hypoxia or alterations of the ionic equilibrium in the inner ear. The rise in the K+ levels in the endolymph and the Na+ levels in the perilymph result in cellular edema and structural damage [88].

Additionally, free radical formation in response to high-intensity noise can lead to vasoconstriction and reperfusion of cochlear cells, with subsequent cell death [81,89,90]. The calcium–magnesium ratio plays a vital role in controlling membrane permeability, voltage-dependent calcium and potassium channels, and polarization. Magnesium is also a potent vasodilator, co-operatively binds with potassium, and has a calcium channel blocker mimetic effect. A decrease in magnesium results in increased membrane permeability, an influx of calcium and sodium into the hair cell, and an efflux of potassium via passive diffusion [91]. In animal models, magnesium deficiency has been reported to exacerbate cochlear trauma in response to noise [92]. A sustained rise in intracellular calcium can deplete cell energy and can also lead to eventual cell death [93,94,95]. Additionally, low magnesium states have been associated with increased catecholamines and prostaglandins inducing vasoconstriction [93,96,97,98,99]. Thus, a decrease in magnesium and an increase in calcium-magnesium ratio can increase blood viscosity, reduce cochlear blood flow, and exacerbate the vasoconstrictor effects of acoustic trauma.

#### 3.3.3. Inflammation

Neuroinflammation, a critical component of maintaining homeostasis in the central and peripheral nervous system, has been implicated in a wide range of pathological processes, including NIHL. There is extensive evidence showing that pro-inflammatory cytokines such as tumor necrosis factor-alpha (TNF-α), interleukins, and chemokines (i.e., CCL2) are induced in the mammalian cochlea after noise trauma [100,101,102,103,104]. Additionally, noise exposure results in the recruitment of inflammatory cells such as macrophages to the cochlea [105,106,107,108]. It is currently unclear whether these inflammatory processes cause or exacerbate the threshold shifts associated with NIHL, although some of these molecules have demonstrated ototoxicity. For example, perfusion of TNF-α into the cochlea of guinea pigs resulted in synaptic degeneration and reduced auditory nerve compound action potentials, which could be protected against by blocking TNF-α [109].

#### 3.3.4. Excitotoxicity and Synaptopathy

Louder sounds cause more hair cells to depolarize, causing greater release of glutamate by IHC ribbon synapses and increased motility of OHC. Noise exposure causes a rapid reduction in the number of IHC ribbon synapses, which is irreversible after neuropathic noise exposure [34,110,111]. Furthermore, excessive release of glutamate leads to glutamate excitotoxicity, characterized by an increased influx of ions in the postsynaptic cochlear nerve terminals leading to swelling of the postsynaptic cell bodies and dendrites [112]. This overstimulation can cause the loss of spiral ganglion cells, even after a delay of several months, and the damage can progress over years [113]. The long-term consequences of primary neuronal loss are progressive and present despite the recovery of threshold sensitivity [34]. Indeed, long-term noise exposure, even beneath the levels to induce cochlea trauma, can result in increased spontaneous firing rates and reorganization of cortical tonotopic maps in mammals [114,115,116,117]. The most common cause of cochlear damage leading to deafferentation is environmental noise overexposure [118,119,120].

Ribbon synapses and the peripheral dendrites of afferent neurons can be damaged by noise levels that are not as high as those required to effectuate threshold shifts [34,121,122]. This deafferentation of the IHC (no contact between them and the dendrites) is termed “synaptopathy” and may not result in detectable threshold shifts. However, as fewer neurons connect to the hair cells, the amplitude growth of neural excitation is reduced, and hearing may be impacted. This phenomenon occurs at noise levels previously thought not to be very harmful. The injury is permanent and not typically detected on an audiogram, hence the term “hidden hearing loss” [34,121,122]. IHCs have two orientation axes: pillar-modiolar and the habenular cuticular. The auditory neurons have high spontaneous rates and lower thresholds on the pillar face, while the modiolar facing neurons have low spontaneous rates and higher thresholds [123,124]. Neurons on the modiolar side of hair cells have smaller receptor patches and more synapses that are more sensitive to noise-induced degeneration compared to the pillar side [125]. In addition, these neurons show a reversible downregulation of protective glutamate receptor expression in peripheral terminals [122]. This spatial distribution of loss of high threshold neurons compared to low threshold neurons might explain the absence of threshold shifts even in the presence of synaptopathy [122].

### 3.4. Additional Negative Effects of Noise on the Inner Ear

#### 3.4.1. Tinnitus

Tinnitus is the conscious perception of sound without an external auditory stimulus, often experienced as ringing or buzzing [126]. Tinnitus is typically self-reported and is primarily subjective [126]. Nevertheless, chronic tinnitus with or without hearing loss has been associated with low-level DPOAEs [127] and low-level transiently evoked otoacoustic emissions [128] compared with people without tinnitus and the same hearing level. Exposure to loud occupational, leisure-time, and firearm noise has been associated with higher prevalence of tinnitus and more frequent symptoms [118,120,129]. Although the precise mechanism of tinnitus remains an area of active research, cochlear deafferentation is thought to play a role. The most common cause of cochlear damage leading to deafferentation is environmental noise overexposure [118,119,120]. Similar to NIHL, neuroinflammation in the auditory cortex may also contribute to tinnitus, with elevated pro-inflammatory cytokines and microglial activation demonstrated in murine models [130,131].

Hearing loss is present in approximately 60% of people with tinnitus [132,133], suggestive of similar pathological processes after harmful noise exposure. The variability between tinnitus and hearing loss in noise-exposed subjects can be due to the differential vulnerability of cochlear and central components to the duration and intensity of noise exposure. A retrospective study of 531 patients with chronic tinnitus found that 83% had a high-frequency hearing loss corresponding to NIHL, and the degree of hearing loss was positively correlated with tinnitus loudness [7]. Similarly, several studies have reported that high proportions (i.e., up to 80%) of military personnel with NIHL also have tinnitus [7,134]. Additionally, the degree of hearing loss positively correlated with the two subscales (“intrusiveness” and “auditory perceptional difficulties”) of the Tinnitus Questionnaire [7]. A study of acute acoustic trauma in the Finnish armed forces found that 47% of personnel reported hearing impairment, and 94% reported tinnitus immediately following acute acoustic trauma, which persisted until discharge in 45% of cases [135]. In addition, otoacoustic emissions were better predictors of tinnitus persistence in a cohort of French military personnel than hearing thresholds alone as early as 24 h after an acute acoustic trauma [136].

#### 3.4.2. Vestibular Dysfunction

The vestibular system consists of the utricle and saccule, which detect gravitational forces and horizontal and vertical plane movements, respectively, as well as three perpendicular semicircular canals [137]. The primary function of the vestibular system is the maintenance of gaze and postural stability, informing head position and spatial orientation, which are crucial to balance [137]. Movement of the endolymph in the semicircular canals during head rotation corresponds to the plane of rotation. Endolymph flows into the ampulla, an expansion of the semicircular canal containing mechanotransducing hair cells, causing deflection of the stereocilia and the release of neurotransmitters that send information about the plane of movement to the brain.

The vestibular labyrinth Is in proximity to and interconnected with the auditory system, and the fluids in the vestibular system have a degree of patency with the cochlea [138]. The cochlea and vestibular hair cells have similar ultrastructure; the balance and auditory receptors share the membranous labyrinth and a common arterial blood supply. These factors increase the likelihood of vestibular trauma in concurrence with NIHL [138]. Similar to the cochlea, vestibular end organ damage by noise can occur via direct mechanical destruction, metabolic decompensation with sensory degeneration, excitotoxicity, and free radical damage [28,139]. Indeed, neurophysiological studies have shown that, similar to the cochlea, the vestibular organs, particularly the saccule and utricle, are vulnerable to noise [140,141,142]. Accordingly, multiple studies have reported associations between NIHL and vestibular dysfunction or balance disorders such as vertigo, oscillopsia, postural instability, and/or motion intolerance [138,143,144,145,146,147].

Neurophysiological studies have measured the effects of noise on the vestibular system via vestibular evoked myogenic potentials (VEMPs) [28,148,149]. Vestibular afferents give rise to VEMPs, which are short-latency myogenic potentials in response to air-conducted sound or bone-conducted vibration. Cervical VEMPs are a measure of the sacculocollic pathway, and ocular VEMPs are a measure of utricular/superior vestibular nerve function [28,148,149]. A cross-sectional observational study of 43 military personnel with bilateral asymmetric hearing loss found that the severity of NIHL was associated with cervical VEMP, suggesting that the sacculocollic pathway is susceptible to noise damage [150]. Indeed, there are saccular neurons responsive to sound [151]. Another measure of vestibular function, particularly of otolith function, is the vestibular short-latency evoked potential [149]. Jerk amplitudes are used to elicit far-field potential responses generated by irregular primary vestibular afferent discharges [152,153]. Studies have shown that vestibular damage can occur due to brief exposure to elevated sound levels and sustained exposure to low-frequency continuous sound at more moderate levels [154,155,156,157]. A study of 258 military personnel found that vestibular damage from intense noise exposure was corelated with asymmetric NIHL [138]. Additionally, symptoms of vestibular dysfunction were observed in 11.2% of individuals in the study with symmetrical hearing loss compared to 21% with asymmetrical hearing loss, which could be attributed to compensation by the vestibular system to symmetrical progressive injury [138]. Additionally, individuals with higher levels of hearing loss (pure tone audiometry [PTA] >40 dB) had more abnormal vestibular test results and worse dual-task performance than those without NIHL [158]. However, participants’ reports of imbalance intensity via the Visual Analog Scale were similar, highlighting the need for vestibular evaluation in patients with hearing loss [158].

## 4. Screening and Diagnosis of NIHL

### 4.1. Screening

Screening of NIHL is typically performed via taking a patient’s history of noise exposure and performing audiograms, and clinicians will distinguish sensorineural from conductive hearing loss. However, it is often difficult to precisely quantify the noise exposure of individuals, and the type of exposure may vary (i.e., intense blast noise, cumulative noise, single tones, or wide-spectrum noise), leading to different patterns of damage. Thus, other tests may include the measurement of distortion product otoacoustic emissions (DPOAE), speech-in-noise testing, and auditory brainstem response (ABR), as detailed below. Each of these methods have notable strengths and limitations. Clinical assessments of signs of tinnitus and vestibular dysfunction may also be included if appropriate.

#### 4.1.1. Audiograms

PTA, which measures the lowest intensity at which sound can still be heard across a range of audible frequencies (hearing threshold), is performed as a baseline evaluation test for hearing loss [159]. Air and bone conduction tests are typically performed over a range of frequencies from 125 to 8000 Hz [160]. NIHL produces characteristics audiometric signatures, where hearing is normal from low to mid frequencies, and there is a sudden drop past 3000 Hz, most pronounced at 4000 Hz, and a slight recovery in higher frequencies (Figure 4) [161]. Multiple factors contribute to this characteristic pattern, including the length and volume of the outer ear canal and the outer ear resonant frequency [60,161,162]. As the duration of noise exposure increases, the notch deepens to involve both higher and lower frequencies.

A 10 dB confirmed threshold shift from the baseline PTA at 2000, 3000, and 4000 Hz is the metric used by the US Occupational Safety and Health Administration (OSHA) to determine a standard threshold shift in cases of occupational hearing loss [163]. While not necessarily indicating hearing impairment, this change can be a warning sign of permanent hearing loss [88]. A limitation of using audiograms for NIHL screening is the need to consider natural age-related declines in hearing sensitivity (i.e., age-associated hearing levels). Thus, age correction of audiograms, which take averages of a population, is recommended for accurate comparisons with a non-noise exposed population [164,165]. However, there is considerable variability in the audiometric profiles among professions. Military personnel are exposed to high impulse sounds and show a pattern of hearing loss more significantly at 6 kHz than at 4 kHz [166].

#### 4.1.2. Speech-in-Noise Testing

While PTA testing helps identify hearing loss, it does not account for the ability to discriminate sound in the presence of background noise, a primary presenting complaint of patients with NIHL [167]. PTA is limited in its ability to effectively predict speech perception because it indicates the patient’s access to sound rather than their functional hearing ability [168]. The speech-in-noise test simulates real-world situations by adding background noise or a competing signal in an isolated sound chamber and better represents the processing of speech [169]. This test measures a patient’s speech recognition threshold and responses to suprathreshold speech (i.e., stimuli presented above the detection threshold) [170]. Signal-to-noise ratio (SNR) loss refers to the increase in SNR required by a listener to repeat words, sentences, or words in sentences with 50% accuracy compared to typical performance [171]. Commonly performed tests are Hearing in Noise Tests, Speech Perception in Noise Test-Revised (SPIN-R), the Bamford–Kowal–Bench Speech-In-Noise (BKB-SIN) test, and QuickSIN [172,173]. In the rapid (~2 min) QuickSIN, test sentences are played at a high SNR of +25 dB to a low SNR of 0 dB, and the patient repeats the sentences back to the audiologist [171]. The number of words that are repeated inaccurately is subtracted from the accurate number.

Speech-in-noise testing gives a useful baseline estimation of hearing impairment, helps to infer the degree of benefit from different assistive hearing devices, and allows the quantification of improvement [174]. Speech-in-noise testing can be used to determine the potential utility of hearing aids for a patient because hearing aids are very effective at improving the ability to understand speech in quiet settings but are less effective in noisy settings [175]. This can result in better alignment with patient treatment priorities, and a 2021 systematic review found that patients who underwent speech-in-noise testing had higher measures of hearing aid satisfaction [176]. Further, this test can potentially detect hearing impairments that may not be evident in PTA testing. For example, when comparing ten noise-exposed Royal Air Force aircrew pilots with ten age/sex-matched unexposed administrators, the groups had comparable audiograms, but the exposed group had relatively elevated SIN thresholds, potentially reflecting abnormal retrocochlear processing [177].

#### 4.1.3. Distortion Product Otoacoustic Emissions (DPOAE) Measurement

The OHCs amplify the intensity and sharpen the peak of the traveling sound wave in the cochlea via prestin-mediated elongation and contraction [178]. The resulting nonlinear electromechanical distortion of the sound wave by OHC motility can be measured via the DPAOE testing [179,180]. In addition to the distortion mechanism, otoacoustic emissions are also attributed to the reflection caused by the random scatter of the incoming traveling wave [180,181]. Therefore, the presence of DPOAEs is a marker of normal cochlear function and, specifically, of OHC health [182].

DPOAEs are evoked via a probe placed into the ear canal which emits two sounds at designated intensity levels (L1 and L2) and frequencies (f1 and f2), while a microphone detects the otoacoustic emission [182]. Cochlear stimulation with two pure tones, f1 and f2, results in a family of distortion products mathematically related to the input tones. Of these, the cubic distortion product 2f1–f2 is most commonly used in clinical and research settings [182]. The test is easy to implement and provides quick results [183].

Decreased DPOAE amplitudes are typically detected in older patients, those exposed to noise, and those with cochlear pathologies [184,185,186]. Thus, the DPOAE testing has been used for myriad clinical applications including newborn hearing screening, diagnostic audiological assessment, ototoxicity monitoring, and the study of cochlear mechanics [187]. DPOAE testing may be superior to PTA testing for military-related NIHL screening, as a study identified it as a quicker, easier tool still capable of detecting early cochlear injury from noise [183].

#### 4.1.4. ABR Measurement

ABR, or auditory brainstem evoked potentials, are electrical signals produced by the nervous system within the first 10–15 ms following a transient acoustic stimulus [188]. A typical ABR consists of a sequence of five vertex positive waves with negative throughs [189]. Regarding the origins of component ABR waves, there are slight differences between humans and rodent models. In humans, wave I and II are believed to originate from the distal and proximal auditory nerve fibers, respectively; wave III is generated by the cochlear nucleus; IV reflects activity in the superior olivary complex; and wave V is associated with the lateral lemniscus [190]. In mice, wave II is believed to be generated by the posterior ventral cochlear nucleus; wave III is believed to be generated by the anterior ventral cochlear nuclear and trapezoid body; wave IV is believed to be generated by the superior olivary complex; and wave V is believed to be generated by the lateral lemniscus and inferior colliculus [191,192,193]. There is also some variation between mammalian models, as reported for rats [192,194,195], guinea pigs [196], and cats [197]. The sound intensity is directly proportional to the waves’ amplitude and inversely proportional to latency.

ABR is not routinely used in clinical screening of NIHL but does have expanded applications for neurodiagnostic testing, intraoperative monitoring, hearing screening/audiometry, and neurophysiological research [193]. In animal models, noise-induced synaptopathy results in a permanent decrease in the supra-threshold growth of ABR wave I amplitude despite full recovery of otoacoustic emission amplitudes and ABR thresholds [110]. A similar pattern of reduced suprathreshold ABR wave I amplitude has been observed in humans exposed to noise, although synaptopathy has not yet been confirmed [198] and is termed “hidden hearing loss” because these functional deficits are hidden by otherwise normal ABR thresholds [63,110]. However, some studies have shown a lack of sensitivity of ABR to noise exposure, which makes wave 1 amplitude a less ideal measure of cochlear synaptopathy for individuals with NIHL [56,170]. These findings underscore the importance of incorporating suprathreshold audiological testing when screening for NIHL to accurately understand a patient’s functional hearing ability. 

### 4.2. Diagnosis

The UK utilizes Coles, Lutman, and Buffin diagnostic guidelines, which identify NIHL by a notch or downward bulge in the frequency range 3–6 kHz during PTA testing [199]. Additional requirements for diagnosing NIHL per these guidelines are high-frequency hearing impairment and a potentially hazardous amount of noise exposure. Four modifying factors which also need consideration include (1) the clinical picture (i.e., the mode, nature, and age of onset, symptom progression, and use of hearing amplification devices), (2) compatibility with age and noise exposure, (3) Robinson’s criteria for other causation [200], and (4) complications such as symptom asymmetry, mixed disorder, and conductive hearing impairment. Modifications have been proposed to account for age-related hearing loss and to delineate the effect of age from NIHL [199]. The US utilizes similar criteria put forth by the American Medical Association in 1979 [201], which were validated in 2011 among 1001 patients via audiometric testing and questionnaires [202]. Modified diagnostic criteria have been proposed to quantify hearing loss thresholds of military-related NIHL by comparing individuals’ hearing thresholds to those of non-noise-exposed individuals [166,171].

## 5. Prevention and Management of NIHL

### 5.1. Prevention

NIHL is mostly preventable, and tangible steps to reduce the burden of the disorder can be taken via the implementation of educational programs, regulation, and legislation to raise awareness and pre-emptively mitigate the damage caused by noise. In the US, the 1972 Noise Control Act established federal noise emission standards for commercial products and required that the public be provided information about noise emission levels and ways of reducing them [203]. Two US governmental departments—OSHA and the National Institute for Occupational Safety & Health (NIOSH)—have made recommendations for the permissible noise limit (PEL) of workplace noise exposure based on the average time a worker is exposed [204,205]. Daily noise dose is expressed as a percentage, per occupational standards, taking duration, sound exposure level, and course of exposure into account. For example, reaching 100% of a worker’s daily noise dose could be expressed as 85 dBA per NIOSH and 90 dBA per OSHA over a shift of 8 h. The course of exposure is cut when there is an increase in noise levels [206] (Table 1). Additionally, OSHA regulates that employers must provide hearing protection if employees are exposed to noise over the permissible exposure limit of 90 dB over an eight-hour time-weighted average [207]. Arenas et al. compared the occupational noise exposure levels in Latin America, the US, and Canada and found that 81% of the countries have a PEL of 85 dBA and that the majority of the countries limit impulsive noise exposure to a peak unweighted sound pressure level of 140 dB [208]. However, there were no established regulations in 27% of the countries, potentially exposing millions of people to NIHL.

Environmental noise level is conventionally measured via a sound level meter. However, they are expensive for small businesses and require maintenance and calibration, thereby limiting their widespread implementation. However, smartphone applications for this purpose are now available and provide inexpensive alternatives to specialist calibrated sound meters. An evaluation of the reliability of nine applications found the NIOShH Sound Level Meter to be the most accurate [209]. A free, accessible, and reliable app may help increase compliance with legislation and easy monitoring of environmental noise levels [210]. Additionally, in 2009, the European Commission mandated that output levels in new personal audio devices should be set to a standard of 85 dB, allowing users to increase the volume to a maximum of 100 dB. When users raise the volume to maximum level, a message was required to pop up that warns of the potential for hearing loss.

Hearing protective devices (HPDs) such as earmuffs and earplugs play an important role in protection against noise exposure. Plugs need to be inserted to ensure coverage of the entire ear canal’s circumference to provide protection and minimize irritation. Noise Reduction Ratings (NRRs) are calibrations to assess the protection range of HPDs in a single attenuation value (in dB). Because NRRs are derived via laboratory-based testing, they may overestimate the actual protection provided in non-experimental environments. Therefore, the NRR is derated by 50% on a dB scale before estimating exposure protection [211]. A Cochrane systematic review (2017) of interventions to prevent occupational NIHL found evidence that training on the proper insertion of ear plugs significantly reduced short-term noise exposure but called for more studies on the effectiveness of stricter legislation or better use of hearing protection devices [212]. Additionally, a randomized controlled trial showed that effective training of earplug usage led to significant improvement in the efficacy of HPDs in comparison to the usage of a device with higher NRR [213]. HPD fit-testing systems provide a customized fit for increased attenuation; however, they often require special facilities for testing such as a quiet room or audiometric booth or specialized equipment. Compliance in wearing HPDs is a barrier, especially in the military, as they are thought to decrease auditory situational awareness (e.g., sound detection, sound localization, and speech perception) [211,214]. However, novel fit test techniques (i.e., via a smartphone application) may enable better training and monitoring compliance [211].

Finally, clinicians can actively counsel patients at risk of NIHL on hearing protection strategies and the hazards of noise exposure in the workplace or recreationally (i.e., concerts, sports events, gun ranges, etc.). Simple strategies outlined by the US Centers for Disease Control and Prevention include avoiding exposure to excessive noise, turning down the volume on music, moving away from sources of noise, and using HPDs to reduce exposure to safe levels [215].

### 5.2. Clinical Management

There is no cure for NIHL and, to date, no approved pharmacological treatment indicated for its treatment. Although there are currently no clinical practice guidelines specifically for NIHL management, such guidelines exist for sensorineural hearing loss in adults (i.e., from the American Academy of Otolaryngology-Head and Neck Surgery (AAOHNS) [216], American Academy of Audiology [217], and the UK National Institute for Health and Care Excellence [218]) and are applicable. Relevant recommendations include exclusion of conductive hearing loss, audiometric confirmation of sensorineural hearing loss (SNHL) characteristic of noise trauma, and exclusion of retrocochlear pathology in case of asymmetric SNHL based on contrast-enhanced brain MRI or ABR testing. NIHL is clinically managed with hearing aids and/or use of hearing protection during exposure, although if hearing loss worsens, patients may be eligible for cochlear implants [56]. In some cases of acute noise-induced TTS, clinicians may consider the use of intratympanic steroids such as dexamethasone [219,220], although high-quality, long-term efficacy evidence in humans is lacking, and it is not considered for chronic occupational noise exposure. Additionally, the WHO classifies hearing loss into mild, moderate, severe, and profound, listing the typical signs and various recommendations for each level of disability (Box 1). Clinicians can use this classification to educate patients regarding the natural history of NIHL while counseling them on protective measures and the benefits of auditory rehabilitation (i.e., hearing aids or other assistive devices). However, the WHO severity levels are arbitrarily defined, and the system does not address patients with <25 dB hearing level and unilateral hearing loss [221]. Additionally, the speech-in-noise test, which is an effective test for measuring noise interference on speech perception skills, is not considered by the WHO system [222].

Box 1WHO grades of hearing loss. WHO grades from the 1991 working group on prevention of deafness and hearing impairment [223]. Additional comments on the classifications from Olusanya et al. [221]. Abbreviations: dB, decibel; ISO, International Organization for Standardization; WHO, World Health Organization.Grade of impairmentAudiometric ISO valuePerformanceRecommendationComments added to the prior classification0-None≥25 dBNo or very slight hearing problem. Can hear whispers.None20 dB also recommended. People with 15–20 dB levels  may have hearing problems. People with unilateral hearing  loss may have problems even if the better ear is normal.1-Slight26–40 dBCan hear and repeat words spoken  in a normal voice at 1 mCounseling. Hearing aids  may be neededSome difficulty in hearing but can usually hear  normal level of conversation2-Moderate41–60 dBCan hear and repeat words spoken in raised voice at 1 mHearing aids are  usually recommendedNone3-Severe61–80 dBCan hear some words when shouted  into better earHearing aids needed. Otherwise  lip-reading and signing should  be taughtDiscrepancies between pure-tone thresholds and speech  discrimination score should be noted4-Profound≥81 dBUnable to hear and understand even  a shouted voiceHearing aids may help understanding  words. Additional rehabilitation needed.  Lip-reading and sometimes signing are  essential.Speech is distorted, the degree depending on the age at  which hearing was lost


## 6. Risk Factors

There are myriad risk factors associated with NIHL, as listed in Box 2 [224,225,226,227,228].

Box 2Common risk factors for NIHL. Abbreviation: NIHL, noise-induced hearing loss.Older age, although all ages are at riskRepeated occupational noise exposure (construction, machine shop/factory, landscaping, mining, agriculture, musician, etc.)Repeated recreational noise exposure (loud music at concerts, loud volume via earphones/earbuds)Intense blast or explosion exposureShooting firearms (recreational or military)Hypertension, smokingLack of hearing protectionExposure to organic solvents, heavy metals, pesticides, asphyxiants


### 6.1. Occupational

Occupational NIHL is the most prevalent occupational disease globally [229], with approximately 7.0 million years lived with disability attributed to occupational noise exposure in 2019 [8]. People employed in construction, manufacturing, mining, agriculture, utility, and transportation industries, military personnel, and musicians are at high risk for NIHL [230]. As discussed above, OSHA considers US workers at increased risk of NIHL if noise exposure is at or above 85 dB, averaged over eight working hours or an eight-hour time-weighted average [231]. According to a report from the NHANES (1999–2004), 22 million US workers (17%) reported exposure to hazardous workplace noise, with 34% of these workers reporting no use of hearing protection devices [232]. Their report identified the highest weighted prevalence of workplace noise exposure for mining (76%) followed by lumber/wood product manufacturing (55%). Other high-risk occupations included repair and maintenance, motor vehicle operators, and construction trades. The position statement from the American College of Occupational and Environmental Medicine (ACOEM) states that occupational and environmental medicine physicians should understand a worker’s history of noise exposure and become proficient in the early detection and prevention of NIHL [233].

### 6.2. Military

Hearing loss is common in the military and is particularly disabling as it relates to both safety of the personnel and that of the nation. Personnel are often exposed to continuous as well as intermittent hazardous levels of noise, including gun fire and blast exposure, and increased hearing loss is associated with solvent exposure and longer service [234,235,236,237]. Combining acoustic and pressure wave energies can result in rupture of the organ of Corti, separation from the basilar membrane, and fracture and displacement of stereocilia. Indeed, as of 2021, tinnitus is the most prevalent service-connected disability for US veterans while hearing loss is the third-most prevalent, affecting 2.5 and 1.4 million veterans, respectively [238]. Individuals with medically disqualifying audiograms or hearing loss at application for service in the US Army and Marine Corp were eight and four times more likely, respectively, to have a hearing loss disability evaluation compared with matched controls [239]. Thus, early identification of enlistees at risk of hearing loss and counseling on hearing protection measures may reduce the burden of NIHL in the military. Currently, the post-exposure protocol for acute acoustic trauma in the US military includes removing the individual from the noise hazard and maintaining them in an effectively quiet environment (ambient levels of ≤70 dB) for 21 days [88,240].

### 6.3. Environmental and Recreational

Environmental factors, including ambient urban noise, also play an important role in the development of NIHL. For example, an observational study of environmental noise in Bangkok, Thailand, found that urban populations experienced greater mean hearing loss than suburban populations and found that among the occupational population in the urban monitoring sites, drivers had the highest risk of traffic-related NIHL [241]. Per a 2011 WHO report, approximately 1.5 million healthy life years are lost due to traffic-related noise in Western Europe alone [242]. Further, recreational noise exposure can contribute to NIHL. A recent systematic review reported that over 1 billion young adults are at risk of permanent, avoidable hearing loss primarily due to unsafe listening practices (i.e., amplified music) [243]. According to the 2021 National Firearms Survey, 81.4 million US adults (31.9 %) own firearms, and recreational shooting is a major cause of non-occupational NIHL [244,245]. Other non-occupational noise sources include chain saws and other power tools, toys, and recreational vehicles such as snowmobiles and motorcycles [9,245].

### 6.4. Genetic

Multiple studies have found that certain genetic and epigenetic characteristics increase the susceptibility to acoustic trauma in animal models of hearing loss [70,246,247,248]. Transgenic mice expressing genes modeling age-related hearing loss, such as *Ahl1* (*Cdh23^735A>G^*) in C57BL/6J mice, are more susceptible to further hearing deterioration caused by noise exposure [247]. Conversely, mice that lack *Ahl1* (i.e., 129/SvEv, Cast/Ei, and MOLF/Ei) are less vulnerable to acoustic injury [249,250,251,252]. Studies of knockout mice have detected pathways involving cochlear structures, oxidative stress, potassium recycling (vital for sensory transduction [85]), and heat shock proteins (HSPs) which increase the susceptibility of the inner ear to NIHL [253]. Some of the mouse genes implicated include *Cdh23* [254], *Pmca2* [255], *Sod1* [83], *Gpx1* [256], *Trpv4* [257], *Vasp* [258], *Hsf1* [259], and *mdx* [249].

Evidence of genetic risk factors in human NIHL is still emerging, although studies have suggested heritability of hearing loss [248,260,261,262]. For example, a study of noise sensitivity among 573 twin pairs in a Finnish cohort demonstrated heritability of 36% overall and of 40% after exclusion of hearing-impaired individuals [248]. Genes in pathways involving oxidative stress, potassium recycling, and HSPs have also been associated with NIHL in humans. Among the oxidative stress pathway genes, polymorphisms of *GSTM1*, *PON2*, *SOD2*, and *CAT* have been associated with NIHL, and *CAT* was independently validated in Swedish and Polish populations [263,264,265]. *GJB1*, *GJB2*, *GJB3*, *GJB4*, *GJB6*, *KCNJ10*, *KCNQ1*, *KCNQ4*, *KCNE1*, and *SLC12A2* are ten genes in the potassium recycling pathway with known or suspected association with syndromic and non-syndromic hearing loss [266]. An analysis of 35 small nuclear polymorphisms in these genes from a Swedish population showed associations between *KCNE1*, *KCNQ1*, and *KCNQ4* with NIHL [266], and a Polish study replicating the study found associations between polymorphisms in *GJB1*, *GJB2*, *GJB4*, *KCNJ10*, and *KCNQ1* with NIHL [267].

HSPs are involved in the synthesis, folding, assembly, and intracellular transport of many other proteins. HPS expression increases during oxidative or other stress, including that caused by noise exposure, which may have a protective effect against cochlear damage via stabilization of stereocilia or enzyme regulation [268,269]. Three polymorphisms in HSP70 genes have been associated with susceptibility to NIHL: *HSP70-1*, *HSP70-2*, and *HSP70-hom* [268,270,271]. An association study with 53 candidate genes in two independent noise-exposed populations identified associations between NIHL and two monogenic genes, the deafness genes protocadherin 15 (*PCDH15*) and myosin 14 (*MYH14*) [272]. Cadherins are responsible for mechanical transduction and form the tip links between hair cells, while myosin regulates cytokinesis, cell motility, and cell polarity in hair cells [43,268]. In addition, noise exposure can result in epigenetic effects (i.e., alternations in gene transcription) via DNA methylation changes [273,274], histone modifications [275,276,277,278], and alterations in non-coding microRNAs (miRNAs) [279,280,281,282].

Finally, the inner ear has many glucocorticoid receptors (GR), which are affected by acoustic trauma or restraint stress at a transcriptional level [283,284,285]. In animal models, acoustic trauma increased endogenous corticosterone and decreased cochlear GR mRNA expression [283,284]. This led to a reduction in nuclear translocation of GR in the spiral ganglion neurons and increased activity of NF-κB. Pretreatment with a glucocorticoid agonist (dexamethasone) resulted in a decreased hearing threshold. Conversely, pretreatment with a glucocorticoid synthesis inhibitor (metyrapone) in combination with a GR antagonist (RU 486) exacerbated auditory threshold shifts (25–60 dB) after acoustic trauma [283].

## 7. Emerging Therapies

### 7.1. Completed and Ongoing Clinical Trials

The development of novel treatments for NIHL is an area of active research, encompassing interventions with anti-inflammatory, antioxidant, anti-excitatory, anti-apoptotic, or other effects. A search of ClinicalTrials.gov on 20 January 2023 identified 21 registered trials of interventional therapies inclusive of NIHL: one active and not recruiting (NCT05086276 (FX-322)), one not yet recruiting (NCT05511753 (acupuncture)), one recruiting (NCT04766853 (steroids)), two enrolling by invitation only (NCT04768569 and NCT04774250 (zonisamide)), one terminated (NCT02903355 (D-methionine)), one withdrawn (NCT02049073 (zonisamide, steroids)), three of unknown status (NCT02779192 (SPI-1005), NCT04482998 (steroids, hyperbaric oxygen therapy [HBOT]), and NCT01727492 (antioxidants)), and 13 completed (NCT02257983 (EPI-743), NCT01444846 (ebselen), NCT00808470 (micronutrients), NCT00552786 (antioxidant), NCT02951715 (zinc), NCT04120116, NCT04601909, and NCT04629664 (FX-322), NCT02259595 (HPN-07, N-acetylcysteine), NCT03834714 (near infrared light), NCT00802425 (AM-111), and NCT03878875 (sound conditioning)) (Appendix A). Of the 12 completed/terminated clinical trials, just five have reported efficacy results via ClinicalTrials.gov or in publications, and all have focused on agents with antioxidant or anti-inflammatory capacity: micronutrients (i.e., vitamins/minerals), D-methionine, N-acetylcysteine, and ebselen (see details in Table 2 and described below). Additional findings have been published from clinical trials conducted outside of the US registry system, with therapies primarily aimed at preventing or attenuating TTS due to noise exposure (Table 3).

### 7.2. Antioxidant Therapy

Beta carotene, vitamins B, C, and E, zinc, and magnesium have antioxidant properties—particularly when combined—and have been shown to reduce vasoconstriction, cochlear cell death, and hearing loss in animal models when administered prior to noise exposure [89,286,287,288,289]. In humans, a 2021 systematic review concluded that certain micronutrients demonstrated protective effects for NIHL but that the results were not consistent [290]. A randomized Phase 2 trial (NCT00808470) of micronutrients (beta carotene, vitamins C and E, and magnesium) administered to young adults prior to exposure to 4 h of loud music reported that it was not superior to placebo in preventing TTS at any frequency level [291]. However, oral magnesium significantly reduced the magnitude of noise-induced TTS and PTS compared with placebo among an Israeli military personnel [292,293]. Similarly, separate prospective randomized trials demonstrated that vitamin B12 [294] and vitamin E (plus inhaled carbogen) [295] significantly attenuated noise-induced TTS compared with placebo. However, in an open-label study of oral zinc (NCT02951715) among adults with NIHL-related tinnitus, treatment improved tinnitus symptoms but not hearing thresholds from baseline after 2 months [296].

N-acetylcysteine is a thiol which supplies cysteine for intracellular glutathione synthesis. It acts as a free radical scavenger of hydrogen peroxide and hydrogen radicals to promote detoxification of ROS and RNS [297,298,299]. A randomized Phase 2 crossover trial (NCT00552786) of N-acetylcysteine administered to male Taiwanese workers exposed to loud occupational noise reported that it significantly reduced the magnitude of TTS from baseline (measured with PTA) compared with placebo among some workers [300]. Specifically, the benefit was limited to those without polymorphisms in the genes *GSTM1* or *GSTT1*, which code for glutathione *S*-transferases (mu and theta, respectively) involved in cellular detoxification and nullification of carcinogens. Thus, antioxidant therapy with N-acetylcysteine may help to reduce the magnitude of noise-induced TTS among certain populations, although individuals’ genetic profiles influence the effect. D-methionine is the active form of methionine, an essential amino acid needed for tissue repair and selenium/zinc absorption. A randomized, double-blind Phase 3 trial (NCT02903355) of D-methionine administered prior to and during firearms training among US drill sergeant trainees (18 total days) reported in ClinicalTrials.gov that it was not superior to placebo in preventing hearing threshold shift. A separate Phase II-like trial of N-acetylcysteine conducted among a similar US Marines Corp population prior to weapons training also failed to demonstrate superiority of treatment to placebo in preventing noise-induced TTS [301].

Sound conditioning, or exposure to low-level noise prior to intense noise, is also aimed at increasing antioxidant activity in the inner ear [302]. This benign stimulus is hypothesized to “prime” the stria vascularis and organ of Corti to produce enzymes (i.e., glutathione reductase, gamma-glutamyl cysteine synthetase, and catalase) protective from noise-induced free radical damage [303]. A completed prospective clinical trial (NCT03878875) examined whether sound conditioning could attenuate TTS and tinnitus among young adults in the UK, although no results have been posted or published. To date, only studies in animal models have demonstrated any benefit from sound conditioning [304,305].

### 7.3. Anti-Inflammatory Therapy

Ebselen is a synthetic selenium-containing molecule that has anti-inflammatory properties and has been assessed as a treatment for various forms of SNHL [306]. It mimics glutathione peroxidase, the primary antioxidant enzyme in the cochlea and which decreases in activity following noise or ototoxic injury, and activates the Keap1-Nrf2 cytoprotective pathway [307,308]. A Phase 2 randomized clinical trial (NCT01444846) among 83 normal-hearing adults compared a placebo with 200, 400, or 600 mg ebselen administered 2 days prior to and 2 days following a calibrated noise exposure challenge [309]. The results indicated that the mean TTS at 4 kHz was significantly reduced (by 68%) in the group receiving 400 mg ebselen compared with placebo.

**Table 2 jcm-12-02347-t002:** Completed clinical trials of interventions for NIHL registered at ClinicalTrials.gov with reported results.

ClinicalTrial.gov ID/Name	Intervention	Study Design	Population, N	Active Arm, N	Comparator Arm, N	Primary Outcome	Secondary Outcomes	Results	Publication	Sponsor
NCT02903355: Phase 3 Clinical Trial: D-methionine to Reduce Noise-Induced Hearing Loss (NIHL)	D-methionine to prevent NIHL or tinnitus in a military population	Phase 3 RCT	US drill sergeant instructor trainees (21–45 y) receiving weapons training, N = 266	18 days of oral D-methionine, n = 124	Placebo, n = 142	ASHA shift from baseline to day 29–36 post-drug (day 15–22 post-noise exposure)	Change in DOEHRS-HC, EWS STS, THI	D- methionine was not superior to placebo on any outcome in the interim analysis	None	Metamor, Inc., USA [discontinued development]
NCT00808470: Micronutrients to Prevent Noise-induced Hearing Loss	Micronutrients to reduce magnitude of TTS from 4 h of loud music in young adults	Phase 2 RCT	Adults (18–35 y) with normal hearing, N = 70	4 days of oral beta carotene, vitamins C and E, magnesium, n = 35	Placebo, n = 35	Mean threshold shift at 4 kHz in both ears 15 min post-music vs. baseline	Threshold shift at 0.25, 0.5, 1, 2, 3, 6, and 8 kHz post-music; change in DPOAE amplitude, PMTF thresholds, and tinnitus measures	Micronutrients were not superior to placebo on any outcome in the final analysis	LePrell et al., 2016 [291]	University of Michigan, USA
NCT00552786: Antioxidation Medication for Noise-induced Hearing Loss	NAC to prevent TTS in workers exposed to noise in Taiwan	Randomized Phase 2 crossover	Males (25–65 y) with known work noise exposure, N = 53	14 days of 1200 mg NAC	Glucose capsule	Threshold shift from baseline measured with PTA, four time-spaced assessments	Threshold shift from baseline measured with DPOAE, four time-spaced assessments	NAC significantly reduced TTS among men without any *GSTIM1* or *GSTT1* polymorphisms	Lin et al. (2010) [300]	National Taiwan University Hospital, Taiwan
NCT02951715: Improvement of Tinnitus After Oral Zinc on Patients With Noise-induced Hearing Loss	Zine to improve NIHL and tinnitus symptoms	Open label, single arm	Adults with confirmed NIHL-related tinnitus, N = 20	2 months of 40 mg oral zinc daily	None	THI change from baseline	Threshold shift, speech discrimination, DPOAE, tinnitus pitch/loudness from baseline	Zinc did not significantly improve hearing threshold but did improve THI score from baseline	Yeh et al. (2019) [296]	Chang Gung Memorial Hospital, China
NCT01444846: Otoprotection With SPI-1005 for Prevention of Temporary Auditory Threshold Shift	Ebselen to prevent TTS	Phase 2 RCT	Adults (18–31 y) with normal hearing, N = 83	4 days of oral 200, 400, or 600 mg ebselen, total n = 63	Placebo capsule, n = 20	Reduction in TTS from baseline (15 min post-sound exposure) at 4 kHz	N/A	Significant reduction in TTS at 4 kHz in 400 mg ebselen vs. placebo groups	Kil et al. (2017) [309]	Sound Pharmaceuticals, Inc., USA

Abbreviations: ASHA, American Speech-Language-Hearing Association; dB, decibel; DOEHRS-HC, Defense Occupational and Environmental Health Readiness System-Hearing Conservation; DPOAE, distortion product otoacoustic emissions; EWS, early warning shift; h, hour; kHz, kilohertz; min, minutes; NAC, N-acetylcysteine; NIHL, noise-induced hearing loss; PMTF, psycho-acoustical modulation transfer function; RCT, randomized controlled trial; STS, standard threshold shift; THI, Tinnitus Handicap Inventory; TTS, temporary threshold shift; US, United States; y, year.

Clinical trials have been conducted to assess HBOT and corticosteroids as treatments for NIHL, primarily focusing on acute acoustic trauma (AAT) (Table 3). Treatment with corticosteroids has been proposed to reduce the inflammation in the inner ear associated with AAT and with intratympanic administration—clinically achieved via laser-assisted myringotomy—resulting in higher perilymph concentrations than systemic administration in animal models [240]. However, the therapeutic benefits are largely predicated on very rapid treatment after NIHL onset. A randomized trial of intratympanic plus systemic or just systemic corticosteroids within 3 days of NIHL onset reported that Chinese patients who received intratympanic delivery had significantly better improvement in audiological outcomes and word recognition than the systemic group, although individuals’ outcomes varied widely [219]. A prospective trial assessed the timing of intravenous corticosteroids and piracetam—a GABA derivative sold outside the US as a nootropic or possible anti-convulsive—as early treatment for AAT among Greek military personnel [310]. The results indicated that patients who were treated within 1 h post-noise exposure achieved the best audiological recovery compared with patients treated 16 or 24+ h later, although there was no control group. However, a case-control study of oral corticosteroids after AAT reported that Israeli miliary personnel treated with <24 h of exposure had significantly better hearing outcomes than those who received no treatment [311].

Hyperbaric oxygen has been previously used to counteract cochlear hypoxia in sudden idiopathic hearing loss [312], including in concert with intratympanic steroids [313]. In the context of NIHL, several retrospective studies have noted that HBOT with or without steroid therapy improved hearing recovery after AAT, although the benefits have been limited and inconsistent [314,315,316,317]. Additionally, a prospective randomized clinical trial examined three regimens of HBOT and/or corticosteroids and piracetam (i.e., medical therapy), following AAT among Belgian soldiers: (1) 10 days oral medical therapy, (2) intensive (twice daily for 3 days then once daily for 7 days) HBOT plus 5 days intravenous then oral medical therapy, or (3) 10 days once-daily HBOT plus oral medical therapy [318]. The results indicated that the audiological outcomes (10 days post-therapy) were superior in the groups that received HBOT versus oral medical therapy alone, although it was not clear if one HBOT regimen was superior to the other. Similar to corticosteroid therapy, the delivery of HBOT treatment very soon after AAT appears crucial to realize benefits [319].

**Table 3 jcm-12-02347-t003:** Additional studies of therapies to prevent or attenuate NIHL (conducted outside of the US).

Study Nation	Author, Year	Intervention	Study Design	Population, N	Active Arm, N	Comparator arm, N	Primary Outcome	Results
Following AAT (attenuation)								
Greece	Psillas et al. (2008) [310]	Prednisone and piracetam	Randomized cohort study	Male soldiers with firearms-related AAT, N = 52	Therapy in <1 h, n = 20	Therapy in >1–<16 h, n = 17 Therapy in 24+ h, n = 15	Complete or partial recovery in 1 month	Significantly higher recovery rate (65%) and better final threshold shifts if treatment at <1 h vs. with delayed treatment (13–24% recovery)
Belgium	Lafère et al. (2010) [318]	HBOT with or without methyl-prednisolone and piracetam (medical therapy)	Cohort study	Belgian soldiers with firearms-related AAT, N = 68	HBOT + oral medical therapy, n = 17 HBOT + IV then oral medical therapy, n = 32	10 days oral medical therapy, N = 17	Average hearing gain and average residual hearing loss 10 days post-treatment vs. baseline pre-AAT	Both regimens of HBOT + medical therapy was superior to medical therapy alone
China	Zhou et al. (2013) [219]	Intratympanic steroid (methyl-prednisolone) with or without oral steroid	Prospective RCT	Adults recently diagnosed with NIHL, N = 52	Intratympanic + oral steroid 3 days after NIHL onset, n = 27	Intratympanic placebo + oral steroid 3 days after NIHL onset, n = 26	PTA and speech discrimination score change from baseline	Significantly more of the intratympanic group had ≥15 dB improvement in PTA and ≥15% speech discrimination vs. placebo + oral steroid only
Before noise exposure (prophylaxis)								
Israel	Attias et al, (2004) [293]	Mg	Prospective RCT	Males (16–37 y) with normal hearing, N = 20	10 days 122 mg oral Mg, n = 10	Placebo, n = 10	TTS immediately following noise exposure (90 dB for 10 min)	12% of Mg-treated patients experienced TTS ≥ 20 dB, vs. 28% in placebo and no-intake groups; reduced magnitude of TTS was significant for all frequencies between 2 and 8 kHz
Israel	Attias et al, (1994) [292]	Mg	Prospective RCT	Male military recruits exposed to noise during 8 weeks of weapons training, N = 255	167 mg oral Mg twice daily during training, n = 125	Placebo, n = 130	PTS 7–10 days post-exposure vs. baseline	PTS was significantly more common and more severe in the placebo vs. Mg-treated group, and negatively correlated with red blood cell Mg level
Italy	Quaranta et al. (2004) [294]	Vitamin B12	Prospective RCT	Adults (20–30 y) with normal hearing, N = 20	1 mg B12 daily for 7 days, then 5 mg on day 8, n = 10	Placebo, n = 10	TTS following 112 dB 3 kHz noise for 10 min measured 8 days post-treatment	Significant reduction in TTS at 3 at 4 kHz in B12 group vs. control
India	Kapoor et al. (2011) [295]	Vitamin E + inhaled carbogen (5% CO_2_ +95% oxygen)	Prospective RCT	Male industrial workers with exposure to intense occupational noise, N = 40	6 days carbogen, vitamin E only, or combination, total n = 30 (10 each)	Placebo, n = 10	TTS from baseline following 5 h noise exposure (90–113.5 dB)	Combined carbogen + vitamin E reduced TTS by 1.6–5.1 dB across frequencies; vitamin E groups had reduced serum markers of oxidative stress

Abbreviations: AAT, acute acoustic trauma; dB, decibel; h, hour; HBOT, hyperbaric oxygen therapy; IV, intravenous; kHz, kilohertz; Mg, magnesium; min, minutes; NAC, N-acetylcysteine; NIHL, noise-induced hearing loss; PTA, pure tone audiometry; PTS, permanent threshold shift; RCT, randomized controlled trial; TTS, temporary threshold shift; y, year.

## 8. Conclusions

This review of the literature described the etiology of and risk factors for NIHL, as well as approaches for screening, prevention, and treatment of affected patients. We also summarized the evidence from publications of experimental therapies which, although promising for prophylaxis or AAT that can be treated very quickly, have yet to address the needs of patients with long-standing or cumulative NIHL. The impact of NIHL is not limited to decreased quality of life of affected individuals but can extend to lower productivity of the workforce and impaired performance of military personnel. Further, NIHL is associated with a substantial economic burden to governments and society at large, expected to increase as the global population ages and experiences increased costs of NIHL-related healthcare and disability. Thus, the prevention of noise-induced cochlear injury, via the use of hearing protection in loud occupational and recreational settings, remains the cornerstone of reducing the enormous burden of NIHL. Governmental regulations and safety-focused organizations play essential roles in advising the public and industry on safe noise exposure levels and in implementing educational programs. For example, a cost-effectiveness analysis of a military hearing conservation program estimated a lifetime savings of USD $10,657 per person in service-related compensation on implementation of the program [320]. Additional harm reduction may also result from better alignment of occupational noise exposure limits recommended by NIOSH and OSHA. It is encouraging that there is increased awareness of the need for NIHL prevention compared to decades ago, although there remains a large unmet medical need for therapies beyond prophylaxis or AAT.

## Figures and Tables

**Figure 1 jcm-12-02347-f001:**
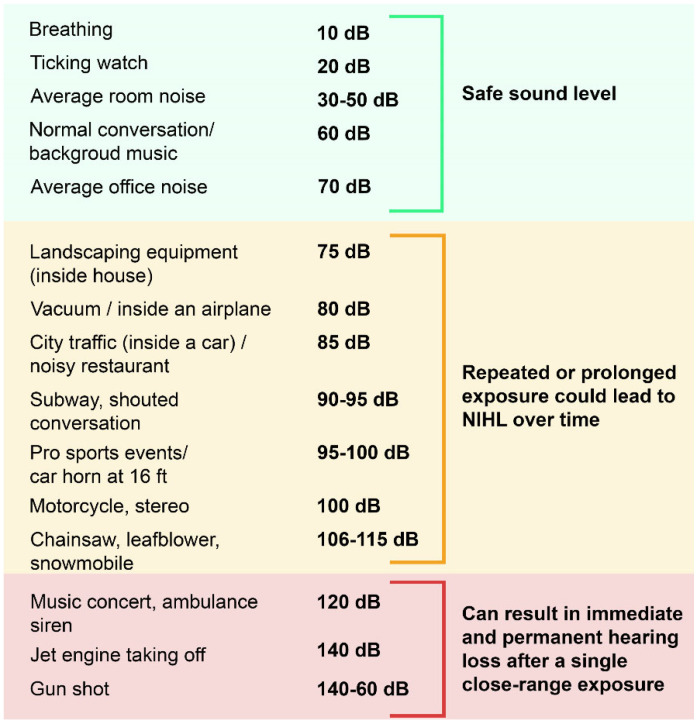
Examples of noise exposure levels in occupational and non-occupational settings. Data from the United States National Institute for Occupational Safety and Health [30] and the Hearing Health Foundation [31].

**Figure 2 jcm-12-02347-f002:**
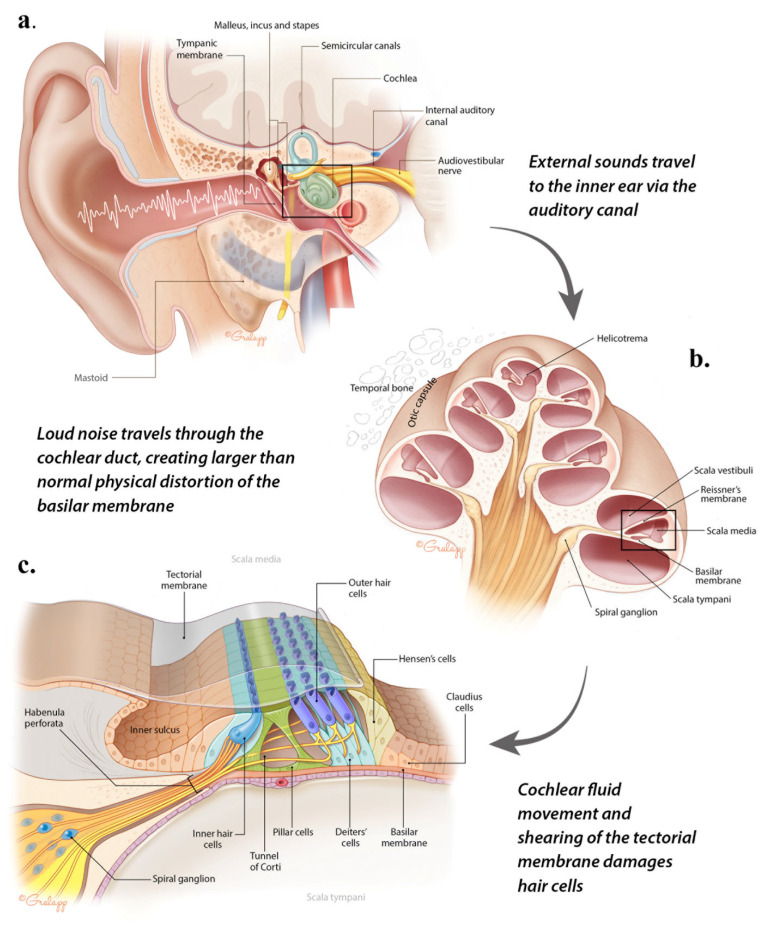
Anatomical structures of the inner ear impacted by NIHL. Three-panel diagram illustrating (**a**) cross section of gross outer, middle, and inner ear anatomy; (**b**) cross-sectional anatomy of the cochlea; (**c**) cellular-level anatomy of the sensory epithelium of the cochlea (organ of Corti). Original illustrations by Christine Gralapp and used with permission.

**Figure 3 jcm-12-02347-f003:**
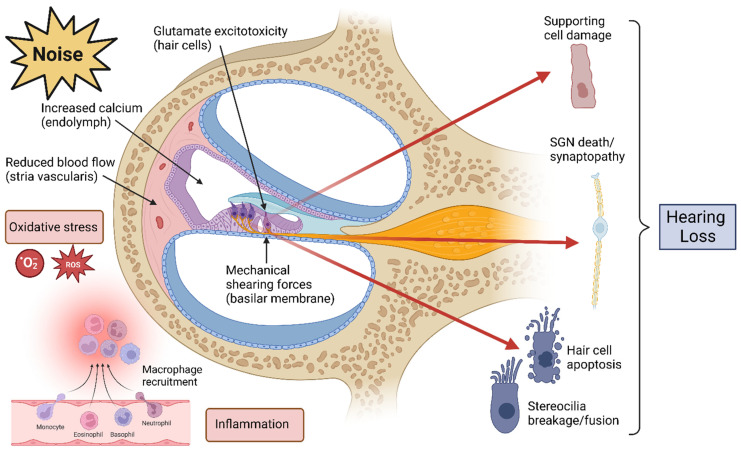
Mechanisms of damage in NIHL. Abbreviations: ROS, reactive oxygen species; SGN, spiral ganglion neurite. Created with BioRender (www.biorender.com; accessed on 7 February 2023).

**Figure 4 jcm-12-02347-f004:**
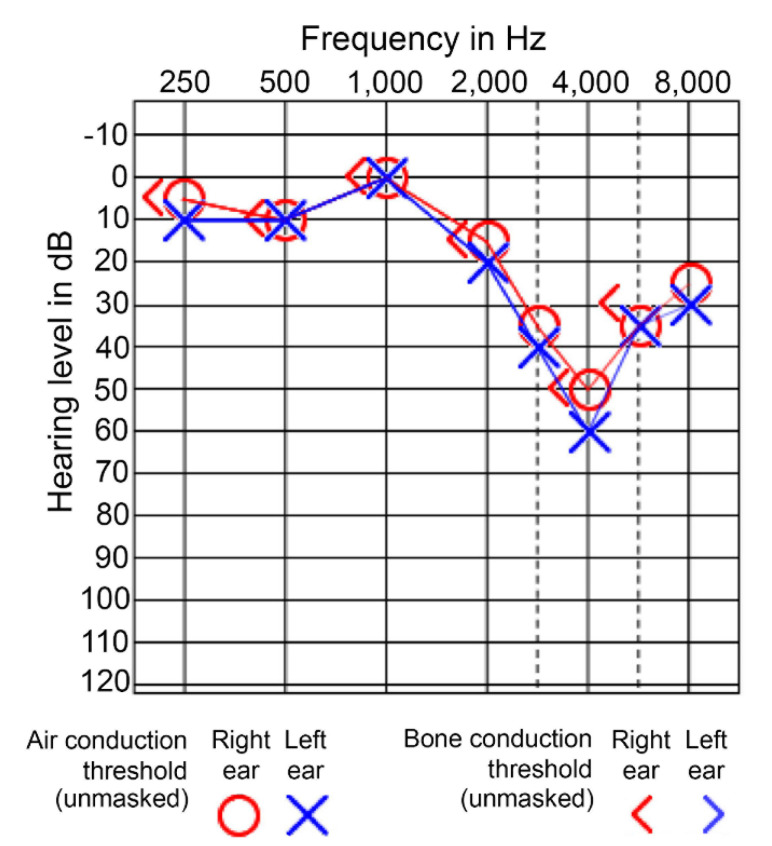
Characteristic audiogram of an individual with NIHL. The signature pattern of hearing loss is focused on 3–4 kHz range. Made with Interactive Audiogram Plotter (https://audprof.com/aud_tools/audiogram/; accessed on 1 February 2023).

**Table 1 jcm-12-02347-t001:** Occupational noise exposure limits recommended by NIOSH and OSHA.

Sound Pressure Level (dB)	Permissible Exposure Time	
	NIOSH	OSHA
120	9 s	7 min 30 s
115	28 s	15 min
112	56 s	22 min 48 s
110	1 min 29 s	30 min
109	1 min 53 s	34 min 12 s
106	3 min 45 s	52 min 12 s
105	4 min 43 s	1 h
103	7 min 30 s	1 h 18 min
100	15 min	2 h
97	30 min	3 h
95	47 min 37 s	4 h
94	1 h	4 h 36 min
91	2 h	7 h
90	2 h 31 min	8 h
88	4 h	10 h 36 min
85	8 h	16 h
82	16 h	24 h 18 min
81	20 h 10 min	27 h 54 min
80	25 h 24 min	32 h

The NIOSH limits represent clinically safe levels, and the OSHA limits are more liberal to account for practical issues in industry [204,205]. Units of time are standardized for ease of comparison. Abbreviations: dB, decibel; OSHA, Occupational Safety and Health Administration; NIOSH, National Institute for Occupational Safety and Health.

## Data Availability

Not applicable.

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
