# Peer review of "Noise-Induced Hearing Loss"

_jcm, 2023, doi:10.3390/jcm12062347_

Round 1
Reviewer 1 Report
Noise-induced hearing loss draws attention in connection with industrialization nowadays. This is a clear, concise, and well-written manuscript. The sections are well-organized and it is easy to read. Epidemiology and pathophysiology sections are very informative and illustrations are very clear. Noise-induced hearing loss was evaluated in all its aspects with sufficient number of references. Limitations of standard audiological testing to diagnose NIHL and importance of supra-threshold testing can be clarified.
Citiationos from Eggermont coud be added especially section 3.1.3 Excitotoxicity and synaptopathy.
There are some witting errors sated as follows - In 4.1.4 ABR section there is difference in font size.
- In title 7.1, there is a misspelling. Written trials and numbers seem complex difficult to read.
Reviewer 2 Report
This review article summarizes various factes of noise induced hearing loss - epidemiology, pathophysiology, Screening/diagnosis, Prevention/management, risk factors and emergent therapies. The article is comprehensive, while having enough depth to serve as an introductory foray into the field of noise induced hearing loss. The organization of the manuscript is logical and clean, and most of the relevant literature in the field are cited. I only have minor comments, detailed below –
Page 2- “ limited access to healthcare and screening tests may leave much of the burden undetected. Further, developing nations may lack governmental guidance or legislation to limit noise exposure, or public education measures to encourage use of hearing protection.” – Please add references
Page 2 – “In Europe, noise was named as one of the main causes of disabling hearing loss affecting over 34.4 million people in 2019, contributing to over 185 billion euros in costs annually” please expand on how the costs were estimated and what this was the cost of.
Page 6-7 – “The mechanisms by which loud noise induces hearing loss includes mechanical damage of cochlear structures, reduction in blood flow, sterile inflammation, and oxidative stress and excitotoxicity due to overstimulation of hair cells and nerves (Fig 3). The loss of hair cells via apoptosis is ultimately the most severe injury and contributes to permanent hearing loss. Noise at specific frequencies can cause discrete areas of hair cell damage which manifest as frequency-specific hearing deficits” – Please add references
Page 7 – “Prolonged duration of exposure to hazardous noise levels or a one-time exposure to high-intensity sound levels can both cause permanent threshold shifts (PTS).” – please add references
Page 9 – “Additionally, free radical formation can lead to vasoconstriction and reperfusion of cochlear cells, with subsequent cell death. The calcium-magnesium ratio plays a vital role in controlling membrane permeability, voltage-dependent calcium and potassium channels, and polarization. Magnesium is also a potent vasodilator, co-operatively binds with potassium, and has a calcium channel blocker mimetic effect. A decrease in magnesium results in increased membrane permeability, an influx of calcium and sodium into the cell, and an efflux of potassium via passive diffusion. A sustained rise in intracellular calcium can deplete cell energy and also lead to eventual cell death [85-87]. Additionally, low magnesium states have been associated with increased catecholamines and prostaglandins inducing vasoconstriction [85,88-91]. Thus, a decrease in magnesium and an increase in calcium-magnesium ratio can increase blood viscosity, reduce cochlear blood flow, and exacerbate the vasoconstrictor effects of acoustic trauma” – Unclear if all these processes happen with noise. If so, please revise for clarity.
Page 13 – “Wave I results from the auditory nerve's compound action potential. Waves II and III are generated by globular cells (wave II) or spherical and globular cells (wave III)” – Wave II is thought to have different generators in humans vs rodent models. While wave II generators are the CN cells in rodents, I believe it is thought to be the proximal eight nerve in humans. Please revise for clarity.
Page 18 – “7. Emerging therapies 7.1. Comp.1leted and ongoing clinical trials” - Typo
